# Differential Responses of *OsMPK*s in IR56 Rice to Two BPH Populations of Different Virulence Levels

**DOI:** 10.3390/ijms19124030

**Published:** 2018-12-13

**Authors:** Satyabrata Nanda, Pin-Jun Wan, San-Yue Yuan, Feng-Xiang Lai, Wei-Xia Wang, Qiang Fu

**Affiliations:** State Key Laboratory of Rice Biology, China National Rice Research Institute, Hangzhou 310006, China; sbn.satyananda@gmail.com (S.N.); yuansanyue@163.com (S.-Y.Y.); laifengxiang@caas.cn (F.-X.L.); weixwang74@126.com (W.-X.W.)

**Keywords:** *Nilaparvata lugens*, MAPKs, rice–BPH interaction, IR56 rice, phytohormones

## Abstract

The conserved mitogen-activated protein kinase (MAPK) cascades play vital roles in plant defense responses against pathogens and insects. In the current study, the expression profiles of 17 *OsMPK*s were determined in the TN1 and IR56 rice varieties under the infestation of brown planthopper (BPH), one of the most destructive hemimetabolous rice pests. The virulent IR56 BPH population (IR56-BPH) and the avirulent TN1 BPH population (TN-BPH) were used to reveal the roles of *OsMPK*s in the compatible (IR56-BPH infested on the TN1 and IR56 rice varieties, and TN1-BPH infested on the TN1 rice variety) and the incompatible (TN1-BPH infested on the IR56 rice variety) interaction. The statistical analysis revealed that rice variety, BPH population type, and infestation period have significant effects on the transcription of *OsMPK*s. Out of these genes, five *OsMPK*s (*OsMPK1*, *OsMPK3*, *OsMPK7*, *OsMPK14*, and *OsMPK16*) were found to exhibit upregulated expression only during incompatible interaction. Six *OsMPK*s (*OsMPK4*, *OsMPK5*, *OsMPK8*, *OsMPK9*, *OsMPK12*, and *OsMPK13*) were associated with both incompatible and compatible interactions. The transcription analysis of salicylic acid, jasmonic acid, and ethylene phytohormone signaling genes revealed their roles during the rice–BPH interactions. The upregulated expression of *OsC4H*, *OsCHS*, and *OsCHI* in the incompatible interaction implied the potential defense regulatory roles of phenylpropanoids. In both varieties, the elevated transcript accumulations of *OsGST* and *OsSOD*, and the increased enzyme activities of POD, SOD, and GST at 1 day post-infestation (dpi), but not at 3 dpi, indicated that reactive oxygen species (ROS) signaling might be an early event in rice–BPH interactions. Furthermore, upregulated transcription of *OsLecRK3* and *OsLecRK4* was found only during an incompatible interaction, suggesting their involvement in the BPH resistance response in the IR56 rice variety. Lastly, based on the findings of this study, we have proposed a model of interactions of IR56 rice with TN1-BPH and IR56-BPH that depicts the resistance and susceptibility reactions, respectively.

## 1. Introduction

The brown planthopper (*Nilaparvata lugens* Stål, hereafter referred to as BPH) is the most harmful pest to rice (*Oryza sativa*), and causes huge crop damage and billions of dollars of economic loss in Asia [1]. It is a typical monophagous vascular feeder that sucks the rice phloem sap, resulting in the wilting and fatal drying of rice plants, a phenomenon known as ‘hopperburn’ [1]. Besides this, BPH causes indirect damage to rice, as it is the carrier of the rice grassy stunt virus and the rice ragged stunt virus [1]. The application of insecticides is the most common practice to restrict and eradicate BPH infestations in rice. In recent years, the abuse of chemical insecticides has resulted in many adversities, such as insecticide resistance development, insect resurgence, the elimination of natural enemies, and environmental hazards. Screening and breeding of cultivars that harbor planthopper resistance genes is considered to be the most desirable and economic strategy for the control and management of BPH [1]. The BPH-resistant sources were first identified in 1967 [2], and then, in 1970, *Bph1* (from Mudgo) and *bph2* (from ASD7), the two BPH resistance genes, were identified in rice [3]. Consequently, these two genes with two other genes, *Bph3* (from Rathu Heenathi) and *bph4* (from Babawee), were extensively used in breeding programs, including breeding of the IR varieties [4,5,6,7]. However, improved cultivars carrying these genes lost their resistance to BPH due to the evolution of new biotypes (populations) [1,8,9,10,11,12]. At present, nine (*Bph3*, *Bph6*, *Bph9*, *Bph14*, *Bph17*, *Bph18*, *Bph26*, *Bph29*, and *Bph32*) of the 34 BPH resistance genes (*Bph/bph*) have been cloned or characterized in rice and its relatives [13,14]. Of them, *Bph3* (a cluster of plasma-membrane-localized lectin receptor kinases, *OsLecRK*s), has been considered for breeding rice cultivars with broad-spectrum and durable insect resistance [7,10]. Recently, we established a new virulent BPH population (IR56-BPH) that could successfully break down *Bph3*-mediated resistance to IR56 rice [9]. After force-feeding over 40 generations, IR56-BPH could break down the IR56 rice’s resistance (standard seedbox screening technique, Grade 7) and exhibited a significantly increased emergence rate, indicating that IR56-BPH had successfully evolved its virulence against IR56 rice [9]. However, the molecular mechanism underlying the interaction between IR56-BPH and its host plant has not been well-elucidated.

The mitogen-activated protein kinase (MAPK) cascade is an evolutionarily conserved pathway in eukaryotes that has three components: an MAPK kinase or MAP3K or MEKK, an MAPK kinase or MAP2K or MEK, and an MAPK [15]. MAPK cascades have been reported to play vital roles in plant defense responses to pathogens and insects [16,17,18]. Although the total numbers and nomenclature of *O. sativa* MAPKs (*OsMPK*s) are ambiguous, only a few studies have revealed the functional roles of *OsMPK*s involved in defense against insects [19,20,21]. For instance, *OsMPK3* (named *OsMPK5* by Rohila and Yang [19]) was reported to be involved in defense against striped stem borer (*Chilo suppressalis*) by modulating jasmonic acid (JA) signaling and herbivore-induced trypsin protease inhibitors levels, but not against BPH [22]. However, *OsMPK3* was also reported to be involved in resistance to BPH and rice blast fungus *Magnaporthe grisea* [23]. To our limited knowledge, these reports are not sufficient for drawing any conclusions about the roles of MAPKs in rice–BPH interactions. Apart from that, phytohormones, including JA, salicylic acid (SA), and ethylene (ET), have been reported to regulate plant defense responses to several biotic stresses [24,25,26]. Furthermore, MAPK cascades and the major plant defense-associated phytohormones, including JA, SA, and ET, have been reported to influence each other; the MAPKs regulate the biosynthesis and accumulation of the phytohormones, while the phytohormones can induce the expression of MAPKs [27]. Moreover, a comprehensive analysis of the modulatory roles of MAPKs in rice’s resistance response to BPH is necessary to find out their molecular mechanisms in rice–BPH interactions.

In the current work, the expression profiling of 17 *OsMPK*s in two contrasting rice varieties, IR56 (a BPH-resistant rice variety containing *Bph3*) and Taichung Native 1 (TN1, a BPH-susceptible rice variety) has been evaluated under the infestation of two BPH populations (an avirulent TN1-BPH, which is incapable of breaking down the resistance of the rice varieties containing *Bph* genes, and a virulent IR56-BPH, which has the capacity to break down *Bph3*-mediated resistance). The rice–BPH interactions have been categorized into compatible and incompatible interactions depending on the virulence of the infested BPH population type on an individual rice variety. The interactions between TN1 rice and TN1-BPH, TN1 rice and IR56-BPH, and IR56 rice and IR56-BPH have been considered as the compatible interactions, whereas the interaction between IR56 rice and TN1-BPH has been considered as the incompatible interaction (Figure 1). Furthermore, the expression profiles of the selected defense response pathway genes, including SA, JA, and ET phytohormone signaling, phenylpropanoid synthesis, and reactive oxygen species (ROS) signaling have been characterized under both compatible and incompatible rice–BPH interactions. Also, the activities of key ROS-related enzymes, including peroxidase (POD) (EC 1.11.1), superoxide dismutase (SOD) (EC 1.15.1.1), catalase (CAT) (EC 1.11.1.6), and glutathione S-transferase (GST) (EC 2.5.1.18), have been evaluated. Additionally, the temporal expression regulation of *OsMPK*s that have been exposed to exogenous phytohormones has been estimated. Lastly, a model of interactions of IR56 rice with TN1-BPH and IR56-BPH that depicts the resistance and susceptibility reactions has been proposed.

## 2. Results

### 2.1. Expression Profiling of OsMPKs in Response to BPH Infestations

The effects of the rice varieties, the BPH populations, and the infestation time periods (0, 1, and 3 days post-infestation, dpi) on the transcription of *OsMPK*s were analyzed by performing a three-way analysis of variance (ANOVA) analysis (Appendix A). The ANOVA analysis revealed that rice varieties, BPH population, dpi, and their interactions (rice varieties × BPH population, rice varieties × dpi, BPH population × dpi, and rice varieties × BPH population × dpi) had no significant effects (*p* >0.05, *F* values are shown in Appendix A) on the expression of *OsMPK6*, *OsMPK10*, *OsMPK11*, *OsMPK15*, and *OsMPK17* (Appendix A). In contrast, all main effects and interaction effects on the expression of *OsMPK3*, *OsMPK5*, *OsMPK7*, *OsMPK12*, *OsMPK13*, *OsMPK14*, and *OsMPK16* (Figure 2A, Appendix A) were significant as revealed by the three-way ANOVA analysis. Out of these significant main and interaction effects, the effects of rice variety on *OsMPK7*, *OsMPK14*, and *OsMPK16* expression, the effects of dpi on *OsMPK3* and *OsMPK5* expression, and the effects of BPH population type on *OsMPK12* and *OsMPK13* expression were higher as compared with the other main and interaction effects. Further, the Tukey′s honestly significant difference (HSD) post-hoc comparisons of these seven genes confirmed three types of upregulated expression patterns. Firstly, *OsMPK3*, *OsMPK7*, *OsMPK14*, and *OsMPK16* were significantly induced in the incompatible interaction, and *OsMPK7* and *OsMPK16* exhibited a time-dependent upregulation. Secondly, elevated mRNA levels of *OsMPK5* were found in both compatible and incompatible interactions. In addition, time-dependent upregulation was found in the IR56-BPH-infested TN1 rice and in the TN1-BPH-infested IR56 rice. Compared with TN1 rice, IR56 rice had higher expression levels of *OsMPK5* after the TN1-BPH infestation, but lower levels of expression after the IR56-BPH infestation. Thirdly, time-independent induced expression of *OsMPK12* and *OsMPK13* was observed in both compatible and incompatible interactions. Compared to those in TN1 rice, higher mRNA levels of *OsMPK12* and *OsMPK13* were found in the incompatible interaction. However, a lower mRNA level of *OsMPK13* was found in IR56 rice at 3 dpi of IR56-BPH infestation.

Furthermore, the expression of *OsMPK1*, *OsMPK2*, *OsMPK4*, *OsMPK8*, and *OsMPK9* was significantly affected by one of the variances and/or their interactions (Figure 2B, Appendix A). The Tukey’s HSD post-hoc comparisons confirmed the downregulated expression of *OsMPK2* in the IR56-BPH-infested IR56 rice (1 dpi), but upregulated expression of *OsMPK1*, *OsMPK4*, *OsMPK8*, and *OsMPK9* in both rice varieties (1 or 3 dpi). Of these induced genes, *OsMPK1* was upregulated in the compatible interaction (IR56 rice infested by TN1-BPH) at 1 dpi. *OsMPK9* was upregulated in both compatible interactions and the incompatible interaction at 3 dpi. Similarly, induced expression of *OsMPK8* was found in both compatible and incompatible interactions at 1 dpi. The expression of *OsMPK4* was time-dependently upregulated in both compatible (TN1 rice infested by IR56-BPH) and incompatible interactions.

Briefly, the expression of five *OsMPK*s (*OsMPK1*, *OsMPK3*, *OsMPK7*, *OsMPK14*, and *OsMPK16*) in the incompatible interaction and six *OsMPKs* (*OsMPK4*, *OsMPK5*, *OsMPK8*, *OsMPK9*, *OsMPK12*, and *OsMPK13*) in both compatible and incompatible interactions was induced under the BPH infestations (Figure 2). These results suggest that the former five *OsMPK*s might be involved in the resistance response to BPH in IR56 rice against TN1-BPH, whereas their failure to achieve upregulated expression under IR56-BPH infestation might have caused the rice’s resistance to break down.

In order to check the effect of tissue injury on the expression of *OsMPK*s in the TN1 and IR56 varieties, wound stress was imposed by mimicking BPH piercing (pricking the rice stems 100 times with a fine needle; no pricked plants were considered as controls). The two-way (rice varieties versus days post-injury, dpi) ANOVA analysis revealed no significant effects of rice varieties and dpi, and no significant effect of rice varieties × dpi interaction, on the expression of *OsMPK1*, *OsMPK2*, *OsMPK9*, *OsMPK10*, *OsMPK12*, *OsMPK14*, *OsMPK15*, and *OsMPK17* (*p* > 0.05, *F*-values for all main and interaction effects are not shown, Appendix A). Furthermore, there was no significant effects of rice varieties, and rice varieties × dpi interaction, on the expression of *OsMPK3*-*5* and *OsMPK11* (*p* > 0.05, *F*-values for rice varieties and interaction effects are not shown, Appendix A), whereas a significant effect of dpi (*OsMPK3*, *F*_1,12_ = 68.44, *p* < 0.001; *OsMPK4*, *F*_1,12_ = 20.42, *p* = 0.002; *OsMPK5*, *F*_1,12_ = 135.39, *p* < 0.001; *OsMPK11*, *F*_1,12_ = 4.38, *p* = 0.037) was observed for the same. A statistical analysis also revealed that no significant effect of rice varieties × dpi interaction (*F*_2,12_ = 3.59, *p* = 0.06), but a significant effect of rice varieties (*F*_1,12_ = 4.76, *p* = 0.049) and dpi (*F*_2,12_ = 5.62, *p* = 0.019), was found on *OsMPK6* expression. Tukey’s HSD post-hoc comparisons confirmed a significant (*p* < 0.01) increase in the mRNA level of *OsMPK3* at day 1 in TN1 rice, and of *OsMPK5* at 1 and 3 dpi in both rice varieties, as compared with the control (without pricking; Appendix A). On the contrary, there were significantly (*p* < 0.01) decreased levels of *OsMPK4* in both rice varieties at 3 dpi, and of *OsMPK6* in TN1 rice at 1 dpi. For four *OsMPK*s (*OsMPK7*, *OsMPK8*, *OsMPK13*, and *OsMPK16*), the effects of rice varieties, dpi, and rice variety × dpi interaction (*p* < 0.01) were statistically significant (Appendix A). Wounding significantly (*p* < 0.05) induced the transcript levels of *OsMPK7* and *OsMPK16* in IR56 rice, *OsMPK8* in TN1 rice, and *OsMPK13* in both rice varieties. After wounding, IR56 rice had higher (*p* < 0.01) mRNA levels of *OsMPK7* and *OsMPK16* than those in TN1 rice, and vice versa for *OsMPK8* and *OsMPK13*. These results revealed that six *OsMPK*s (*OsMPK3*, *OsMPK5*, and *OsMPK13* in both rice varieties, *OsMPK8* in TN1 rice, and *OsMPK7* and *OsMPK16* in IR56 rice) were upregulated in response to wounding in rice, while two *OsMPK*s (*OsMPK4* in both rice varieties and *OsMPK6* in TN1 rice) were downregulated.

### 2.2. Induced Expression of Genes Associated with Phytohormone Signaling and Phenylpropanoid Synthesis under BPH Infestation

Insect herbivory and activation of MAPKs can influence the phytohormone balance and secondary metabolite synthesis in plants [24]. A two-way ANOVA analysis of the expressions of *phenylalanine ammonia lyase 1* (*OsPAL1*), *enhanced disease susceptibility 1* (*OsEDS1*), and *nonexpresser of pathogenesis-related genes 1* (*OsNPR1*), which are involved in SA signaling, revealed the positive role of SA signaling in rice–BPH interactions (Appendix A). In IR56 rice, infestation of TN1-BPH (the incompatible interaction) resulted in significant elevated transcript accumulations of *OsPAL1* (*F*_2,12_ = 11.90, *p* = 0.001; *F*_2,12_ = 99.03, *p* < 0.01) and *OsEDS1* (*F*_2,12_ = 6.99, *p* = 0.009; *F*_2,12_ = 62.12, *p* < 0.01) at both 1 and 3 dpi (Figure 3A), and of *OsNPR1* (*F*_2,12_ = 4.28, *p* = 0.039) at 1 dpi. However, IR56-BPH infestation (a compatible interaction) did not result in any significant (*p* > 0.05, *F* value not shown) induced expression of the three genes. Similarly, in TN1 rice, infestation of TN1-BPH or IR56-BPH (compatible interactions) caused no significant upregulation (*p* > 0.05, *F* value not shown) of *OsPAL1*, *OsEDS1*, and *OsNPR1* at either 1 or 3 dpi (Figure 3A).

The expression of the JA biosynthesis genes *allene oxide synthase 1* (*OsAOS1*), *allene oxide synthase 2* (*OsAOS2*), and *lipoxygenase 1* (*OsLOX1*) was determined in compatible and incompatible interactions. The two-way ANOVA analysis revealed that the three genes were significantly induced in both varieties under BPH infestation (Figure 3B, Appendix A). In both rice varieties, *OsAOS1* (*F*_2,12_ = 7.45, *p* = 0.007), *OsAOS2* (*F*_2,12_ = 39.42, *p* < 0.001), and *OsLOX1* (*F*_2,12_ = 23.05, *p* < 0.01) were found to exhibit significant upregulated expression at 1 dpi under the infestation of TN1-BPH or IR56-BPH; however, no significant induced expression was found at 3 dpi.

The expression of 1-aminocyclopropane-1-carboxylate synthase 2 (OsACS2), ethylene-insensitive gene 2 (OsEIN2), and ethylene-responsive transcription factor 3 (OsERF3), which are ET signaling genes, was determined. A statistical analysis revealed that the three genes were significantly induced during incompatible rice–BPH interactions at both 1 and 3 dpi (Figure 3C, Appendix A). In IR56 rice, infestation of TN1-BPH resulted in a significant increase in the transcription of OsACS2 (F_2,12_ = 7.46, *p* = 0.007; F_2,12_ = 41.62, *p* < 0.01), OsEIN2 (F_2,12_ = 14.36, *p* < 0.01; F_2,12_ = 53.71, *p* < 0.01), and OsERF3 (F_2,12_ = 119.7, *p* < 0.01; F_2,12_ = 276.03, *p* < 0.01) at both 1 and 3 dpi. However, when infested with IR56-BPH, only OsERF3 (F_1,12_ = 539.87, *p* < 0.01) was found to be significantly induced in IR56 rice at 3 dpi, while OsACS2 and OsEIN2 showed no significant induced expression at either 1 or 3 dpi. In contrast, none of these three genes showed any significant transcript accumulations in TN1 rice under infestation of TN1-BPH or IR56-BPH. Briefly, the induced expression of the SA and ET signaling genes during the incompatible, but not during the compatible, rice–BPH interactions suggests that they have positive roles in rice’s resistance responses. Conversely, significant upregulation of JA signaling genes at 1 dpi, but not at 3 dpi, and during both compatible and incompatible rice–BPH interactions, suggests that JA signaling might be involved in an early response to BPH feeding in rice.

The expression of the phenylpropanoid biosynthesis pathway genes *cinnamate-4-hydroxylase* (*OsC4H*), *chalcone synthase* (*OsCHS*), and *chalcone isomerase* (*OsCHI*) showed that all three genes were significantly upregulated during the incompatible rice–BPH interaction. In IR56 rice, *OsC4H* (*F*_2,12_ = 59.92, *p* < 0.01; *F*_2,12_ = 26.14, *p* < 0.01), *OsCHS* (*F =*162.64.7, *p* < 0.01; *F*_2,12_ = 109.95, *p* < 0.01), and *OsCHI* (*F*_2,12_ = 54.08, *p* < 0.01; *F*_2,12_ = 37.38, *p* < 0.01) were found to exhibit elevated expression levels at both 1 and 3 dpi under TN1-BPH infestation (Figure 4B, Appendix A). However, under IR56-BPH infestation (a compatible interaction), *OsCHI* (*F*_2,12_ = 14.64, *p* < 0.01) showed significant induced expression at 3 dpi, whereas the other two genes did not show any upregulation in their expression levels at 1 or 3 dpi. On the other hand, none of these three genes showed any significant induced expression in TN1 rice under the infestation of either TN1-BPH or IR56-BPH. The induced expression of *OsC4H*, *OsCHS*, and *OsCHI* during the incompatible interaction suggests that phenylpropanoids have a positive role in rice–BPH interactions.

### 2.3. Expression Profiling and Enzyme Activity of ROS-Signaling Genes under BPH Infestation

A two-way ANOVA analysis of the expression of two ROS-responsive genes, *superoxide dismutase* (*OsSOD*) and *glutathione S-transferase* (*OsGST*), in the TN1 and IR56 rice varieties under BPH infestation revealed that ROS signaling is associated with rice–BPH interactions as an early response, irrespective of the infested BPH population type. In both TN1 and IR56 rice, infestation of TN1-BPH or IR56-BPH resulted in the significant upregulation of *OsSOD* (*F*_2,12_ = 25.13, *p* < 0.01) and *OsGST* (*F*_2,12_ = 18.95, *p* < 0.01) at 1 dpi, whereas no significant induced expression of both the genes was observed at 3 dpi (Figure 4A, Appendix A). Furthermore, the enzyme activity of POD (*F*_2,12_ = 24.93, *p* < 0.01) and SOD (*F*_2,12_ = 5.93, *p* = 0.016) was found to be significantly increased at 1 dpi in both the TN1 and IR56 rice varieties, irrespective of the infested BPH population type (Figure 5, Appendix A). In TN1 rice, under TN1-BPH or IR56-BPH infestation, both POD and SOD showed no significant change in the enzyme activity at 3 dpi as compared to the control plants. However, in IR56 rice, infestation of either TN1-BPH or IR56-BPH caused a significant decrease in the enzyme activity of SOD (*F*_2,12_ = 65.04, *p* < 0.01) at 3 dpi. Furthermore, in TN1 rice, infestation by both BPH populations caused a significant increase in GST enzyme activity (*F*_2,12_ = 87.80, *p* < 0.01) at 1 dpi. In IR56 rice, the infestation of IR56-BPH caused a significant increased activity of GST (*F*_2,12_ = 133.79, *p* < 0.01) at 1 dpi, while TN1-BPH infestation resulted in no significant change in GST activity. Conversely, at 3 dpi, GST activity in both the TN1 and IR56 rice varieties was significantly decreased (*F*_2,12_ = 83.92, *p* < 0.01) in response to BPH infestation, irrespective of the population type. In addition, the enzyme activity of CAT in TN1 and IR56 rice was found to be significantly downregulated (*F*_2,12_ = 8.42, *p* = 0.005; *F*_2,12_ = 8.03, *p* = 0.006) under TN1-BPH infestation at both 1 and 3 dpi as compared with the control. However, infestation of IR56-BPH caused no significant change in CAT activity in both TN1 and IR56 rice at 1 dpi, while resulting in decreased activity at 3 dpi in TN1 rice. These results suggest that, like JA signaling, ROS might play a role in the early responses of rice to BPH feeding.

### 2.4. Differential Transcription of OsLecRKs in the Two Rice Varieties under BPH Infestation

A separate study in our lab revealed that both IR56 rice and TN1 rice possess three *OsLecRKs* (*OsLecRK1*, *OsLecRK3*, and *OsLecRK4*) out of the four from the *Bph3* gene cluster (unpublished data). The transcription responses of these three *OsLecRKs* in both rice varieties were analyzed under BPH infestation. In IR56 rice, the TN1-BPH infestation (the incompatible interaction) significantly upregulated the expression of *OsLecRK3* (*F*_2,12_ = 51.95, *p* < 0.01; *F*_2,12_ = 56.22, *p* < 0.01) and *OsLecRK4* (*F*_2,12_ = 115.12, *p* < 0.01; *F*_2,12_ = 115.73, *p* < 0.01) at 1 and 3 dpi, whereas no significant upregulation of *OsLecRK1* (*p* > 0.05, *F* value not shown) was observed (Figure 6, Appendix A). On the other hand, infestation with IR56-BPH did not cause any significant induced expression of any of these three genes in IR56 rice either at 1 or 3 dpi. Similarly, in TN1 rice, infestation of TN1-BPH or IR56-BPH (compatible interactions) induced no significant upregulation of the *OsLecRK*s. From the results, it could be suggested that *OsLecRK3* and *OsLecRK4* play an important role in the resistance of IR56 rice to TN1-BPH, but their activation might be hindered by IR56-BPH during the infestation, resulting in a breakdown in the IR56 rice′s resistance.

### 2.5. Expression Modulation of the BPH-Induced OsMPKs in Response to the Exogenous Treatment of Phytohormones

Out of 11 BPH-induced *OsMPK*s, the exogenous treatment of SA resulted in significant activation (*p* < 0.05) of two *OsMPK*s (*OsMPK4* and *OsMPK5*) in both rice varieties, and of four *OsMPK*s (*OsMPK9*, *OsMPK12*, *OsMPK14*, and *OsMPK16*) in IR56 rice (Figure 7A, Appendix A). Similarly, treatment of methyl jasmonate (MeJA) significantly activated three *OsMPK*s (*OsMPK3*, *OsMPK5*, and *OsMPK13*) in both rice varieties, and of four *OsMPK*s (*OsMPK1*, *OsMPK4*, *OsMPK7*, and *OsMPK9*) in IR56 rice (Figure 7B, Appendix A). Likewise, two *OsMPK*s (*OsMPK9* and *OsMPK12*) showed upregulated expression in both rice varieties under exogenous treatment of ethephon, and five *OsMPK*s (*OsMPK4*, *OsMPK7*, *OsMPK13*, *OsMPK14*, and *OsMPK16*) exhibited significant induced expression in IR56 rice (Figure 7C, Appendix A). These results showed that the number of *OsMPK*s triggered by the exogenous treatment of phytohormones was higher in IR56 rice as compared to TN1 rice.

## 3. Discussion

In the present study, the temporal expression profiles of *OsMPK*s in two contrasting rice varieties were determined in response to BPH infestation at varied virulence levels. The *OsMPK* expression revealed that 11 *OsMPK*s (*OsMPK1*, *OsMPK3–5*, *OsMPK7–9*, *OsMPK12–14*, and *OsMPK16*) were upregulated in the incompatible interaction. Of them, upregulation of *OsMPK4*, *OsMPK5*, *OsMPK8*, *OsMPK9*, *OsMPK12*, and *OsMPK13* was also observed in the compatible interactions. Thus, the induced expression of these genes suggested the involvement of multiple *OsMPK*s in response to BPH-feeding. Differential transcription of multiple *OsMPK*s, including *OsMPK5*, *OsMPK12*, *OsMPK13*, and *OsMPK17*, has been reported in rice in response to BPH infestation [28]. Further, four MAPKs (*SIPK* and *WIPK* in tobacco, and *LeMPK1* and *LeMPK2* in tomato) have been reported to be involved in defense responses to the chewing insect *Manduca sexta* by regulating the expression of the downstream defense genes, secondary metabolite genes, and phytohormone signaling [24,29]. Additionally, *OsMPK3*, *OsMPK6*, and seven homologs (*CaMAPK2–3*, *CaMAPK5*, *CaMAPK7–9*, and *CaMAPK15*) in chickpea have been identified to be induced in response to insect infestations [30,31]. Furthermore, nine *OsMPK*s (*OsMPK2*, *OsMPK4–5*, *OsMPK7–8*, *OsMPK12–13*, *OsMPK15*, and *OsMPK17*) were reported to show induced expression after infection of *Magnaporthe grisea*, whereas three *OsMPK*s (*OsMPK5*, *OsMPK13*, and *OsMPK17*) were induced by the treatment of both virulent and avirulent fungal inoculations [32]. Activation of *OsMPK4*, *OsMPK5*, *OsMPK7*, *OsMPK8*, and *OsMPK13* was found to be common between the current study and the experimental results in that of Reyna and Yang [32]. Thus, the activation of *OsMPK5*, *OsMPK13*, and *OsMPK17* might be associated with both insect and pathogen resistance in rice. Moreover, the induced expression of multiple *OsMPK*s in response to the BPH infestation in the current study is in accordance with previously reported studies (Table 1).

The differential transcript patterns of *OsMPK*s during rice–BPH interactions suggests that *OsMPK* expression can be correlated with the infested BPH population types. During the incompatible interaction, a higher number of *OsMPK*s exhibited induced expression as compared to the compatible interactions. These results suggest that, during the compatible interactions, the virulent BPH succeeded in suppressing or breaking down the rice’s resistance, possibly by employing explicit effectors. The presence of proteinaceous substances in BPH saliva suggests that the salivary secretions might serve as a source of effectors to outrun rice defenses. For instance, the salivary protein NlSEF1 with an EF-hand Ca^2+^-binding domain and the endo-β-1,4-glucanase (NlEG1) were reported to suppress rice defenses by manipulating defense signaling, including SA and JA [37,38]. Moreover, the virulence of pathogens and the host resistance responses are usually host-specific or isolate-specific, which function among discrete races of a pathogen and distinct genotypes or cultivars of a plant species [39]. Further, effector-triggered immunity (ETI) and effector-triggered susceptibility (ETS) are host-specific in nature [40]. Thus, the IR56-BPH that had evolved its virulence against IR56 rice might have broken down the rice’s resistance by following a mechanism similar to ETS. In addition, changing the host, i.e., infesting TN1 rice with IR56-BPH, and IR56 rice with TN1-BPH, might have impaired the BPH effector-mediated ETS, and, thus, activated the cross-population responsive *OsMPK*s (*OsMPK4*, *OsMPK8*, and *OsMPK9*) in the rice varieties.

In the current study, the analysis of the transcript accumulations of genes involved in SA, JA, and ET signaling during the rice–BPH interactions suggested that BPH resistance in rice is possibly modulated by interplay between SA, JA, and ET. Significant upregulation of the three SA signaling genes *OsPAL1*, *OsEDS1*, and *OsNPR1* during the incompatible interaction suggested the intrinsic role of SA signaling in BPH resistance responses in rice. *OsPAL1* is a crucial enzyme in the phenylpropanoid pathway via which plants synthesize SA [41]. *OsEDS1* plays a major role in conferring SA-dependent resistance against pathogens, while *OsNPR1* is a key gene involved in SA-mediated systemic acquired resistance (SAR) [42,43]. Similarly, the expression of *OsACS2*, *OsEIN2*, and *OsERF3* was significantly upregulated during the incompatible interaction. *OsEIN2* positively regulates ET signaling and the production of ROS and phytoalexins, whereas *OsERF3* has been reported to act as a central switch in rice to direct plant defense responses in a herbivore-specific manner by controlling many crucial pathways, including MAPK activation and ET, SA, JA, and H_2_O_2_ signaling [44,45]. Also, the activation of the ET signaling pathway was observed in rice under BPH infestation that, in turn, upregulated the transcription of the BPH-induced gene *Bphi008a* [28]. Induced expression of *OsPAL1*, *OsEDS1*, and *OsNPR1*, along with other SA-responsive genes, was observed during the *Bph14*-mediated resistance that triggered the SA signaling pathways in a resistant rice variety [46]. Similarly, expression of *OsACS1*, *OsEIN2*, and *OsERF1* was found to be elevated in a resistant rice under *Meloidogyne graminicola* infection [47]. Conversely, the analysis of the expression of *OsAOS1*, *OsAOS2*, and *OsLOX1* during the rice–BPH interactions revealed that induced activation of JA biosynthesis genes is an early event during rice–BPH interactions and is independent of the BPH population type. These findings from the current work are in accordance with some of the previously reported results, where wounding or herbivory by chewing insects profoundly induced JA biosynthesis, whereas BPH infestation has resulted in either a very weak JA burst or the JA signaling being negatively correlated with BPH resistance in rice [44,48,49]. Furthermore, a cDNA microarray analysis confirmed that BPH infestation in the resistant rice variety RH induced SA biosynthesis genes and SA levels, but not JA biosynthesis genes or intrinsic JA levels [50]. Thus, from our experimental findings, we hypothesize that SA and ET signaling might play an important role in the BPH resistance response in rice.

Elevated expression of the key phenylpropanoid biosynthesis genes during the incompatible rice–BPH interaction suggested the potential role of secondary metabolite production in rice in response to BPH infestation. Moreover, upregulated expression of *OsC4H*, *OsCHS*, and *OsCHI* only during the incompatible interaction suggested a role for phenylpropanoids in BPH resistance. Increased production of secondary metabolites, including phenolamides and p-coumaroylputrescine, has been reported in rice under the infestation of the chewing pests *Spodoptera mauritia* and *Parnara guttata* [51]. Similarly, in response to BPH infestation, upregulation of the *PAL* gene and the subsequent production of phenylpropanoids and polyphenols were reported by Liu et al. [52]. The analysis of the expression of ROS-responsive genes and the estimation of the ROS-responsive enzyme activities during the rice–BPH interactions suggested that the generation of ROS is an early event in rice–BPH interactions and might be associated with the basal defense response. The upregulated expression of *OsGST* and *OsSOD* and the increased enzyme activity of POD and SOD at 1 dpi, but not at 3 dpi, during both compatible and incompatible rice–BPH interactions indicate that ROS accumulation is an early event in rice–BPH interactions and not BPH-population-specific in rice.

In the current work, *OsLecRK3* and *OsLecRK4* exhibited induced expression only during incompatible rice–BPH interactions, indicating their functionality in BPH resistance in IR56 rice. Similar expression patterns were reported for *OsLecRK1-3* in RH rice under BPH infestation [7]. Further, the LecRKs belong to a specific pattern recognition receptor (PRR) family and are upstream of MAPKs [53]. In our study, the induced expression of *OsLecRK*s during the incompatible rice–BPH interaction suggested potential *OsLecRK* and *OsMPK* interactions in the BPH resistance response. In *Arabidopsis*, *AtLecRK*-*IX2* regulated the activation of *AtMAPK3* and *AtMPK6* in response to flg22 treatment [53]. In pepper, *CaLecRK*-*S.5*, which confers broad-spectrum resistance, regulated the activation of *CaMK1* and *CaMK2* [54]. Thus, we propose that *OsLecRK*s possibly regulate rice’s resistance against BPH infestation by activating downstream MAPKs. However, more in-depth experiments, including interaction studies between *OsLecRKs* and *OsMPKs*, are needed to confirm their association with respect to BPH resistance in rice.

The differential expression of the BPH-induced *OsMPK*s in response to exogenous treatment of signal molecules, including SA, JA, and ET, revealed their potential role in mediating different hormone signals in rice. Two *OsMPK*s (*OsMPK4* and *OsMPK9*) were found to be induced in IR56 rice by the treatment of all three phytohormones, suggesting that they have prospective roles in influencing SA, JA, and ET interplay. Similar results were reported by Reyna and Yang [32], where exogenous treatment of different signaling molecules induced the expression of blast fungus responsive *OsMPK5*, *OsMPK12*, and *OsMPK13*. Plant MAPKs have been reported to regulate multiple phytohormone signaling pathways, and some of them act as the nexus for two or more hormone signaling pathways [55]. *OsMPK3* and *OsMPK6* have been reported to act as the nexus in the interplay of auxin and cytokinin [56]. Moreover, the discrimination in the number of *OsMPK*-induced expressions by the exogenous phytohormone treatments between TN1 rice and IR56 rice might be because additional *OsMPK*s are involved in phytohormone signaling in the resistant rice as compared to the susceptible one. 

In our attempt to elucidate the functionality of the induced *OsMPK*s during the rice–BPH interactions, we performed a chemical inhibition (non-specific and broad-spectrum) of the rice MAPKs by using a mixture of PD98059 and Genistein. PD98059 is a selective inhibitor of MEK1 and MEK2; thus, it can suppress the activation of downstream MEK1- or MEK2-dependent MAPKs [57]. On the other hand, Genistein is a non-specific MAPK inhibitor (kinase inhibitor) that can suppress the activation of MAPKs in plants [58]. Application of the mixture of PD98059 and Genistein resulted in the transient downregulation of all five candidate *OsMPK*s (*OsMPK1*, *OsMPK3*, *OsMPK7*, *OsMPK14*, and *OsMPK16*) (Appendix A). The BPH bioassays with TN1-BPH or IR56-BPH on the inhibitor-mix-treated plants of both TN1 and IR56 rices were found to be non-significant as compared with the controls (untreated) (Appendix A). The possible reason behind this might be because of the transient nature of the chemical inhibition of MAPKs. The use of the inhibitors transiently suppressed the expression of *MAPK*; however, their restored expression in the plants might have conferred resistance to BPH infestation. Thus, targeted mutation of specific *OsMPK*s can further clarify their roles in the rice–BPH interactions. Moreover, repeated treatments (alternative day) of the inhibitor mix to prolong the MAPK inhibition resulted in fatal drying of the mock plants (chemical-treated, no BPH) of both rice varieties. Thus, the results obtained from this experiment were considered to be inconclusive, and loss of function studies have been planned to determine the roles of the candidate *OsMPK* in the rice–BPH interactions.

From the experimental findings in the study, we propose a transcriptional-level model depicting the IR56 rice’s resistance mechanism against TN1-BPH, and the resistance breakdown in IR56 rice by IR56-BPH (Figure 8). In IR56 rice, the activation of *OsLecRK*s in response to TN1-BPH could possibly be achieved by the recognition of the damage-associated molecular patterns (DAMPs), herbivore-associated molecular patterns (HAMPs), or effectors. For instance, oligogalacturonides, oligomers of alpha-1,4-linked galacturonosyl residues, are one of the well-studied DAMPs that are released from plant cell walls upon degradation by insect feeding and can elicit plant defense responses [59]. *OsLecRK*s activation then might induce the downstream *OsMPK*s via production of intermediate factors or signal molecules [60,61]. The activated MAPK cascade might then, in turn, activate the phytohormonal defense’s regulation and other downstream transcriptional reprogramming to confer resistance against BPH. SA induces the deposition of callose in phloem cells and the production of a trypsin inhibitor to discourage BPH feeding, whereas ET results in the production of a green volatile that deters BPH feeding [1,46]. The activation of a secondary metabolite biosynthesis pathway, such as phenylpropanoids, results in the production of important compounds, such as lignin and phytoalexins, which help to mitigate BPH pressure [61]. However, when IR56 rice was infested with IR56-BPH, none of the *OsLecRK* were found to be induced, suggesting that IR56-BPH can suppress the PRR-mediated immune response in rice, possibly by the use of other explicit effectors. The chitin elicitor binding protein (CEBiP)-mediated immunity in rice against *M. oryzae* was suppressed by secreting the effector Secreted LysM Protein1 (Slp1) [62]. Furthermore, infestation of IR56-BPH in IR56 rice successfully suppressed the induced expression of all BPH-responsive *OsMPK*s except *OsMPK5*, which was activated during TN1-BPH infestation. In addition, none of the SA or ET signaling genes were found to be activated during IR56-BPH infestation, and the same was observed for the phenylpropanoid biosynthesis genes. Taken together, these results suggest that IR56-BPH could have broken down the resistance of IR56 rice by suppressing the transcription of many defense-responsive pathways, including the MAPKs. Silencing of *LecRK*s had resulted in a reduced hypersensitive response, secondary metabolite production, and MAPK expression in *Arabidopsis* [54]. On the one hand, incompatible rice can tackle BPH infestation either by the identification of effectors, the initiation of a hypersensitive response, the activation of downstream defense pathways, including MAPKs, phytohormone biosynthesis, or secondary metabolite production, or by a cocktail of these responses. On the other hand, suppressing these plant defense mechanisms by ETS or by a similar mechanism, the virulent BPHs outrun the rice’s resistance in the compatible interactions. Currently, the identification of such effector molecules in IR56-BPH is in progress in our lab, and the results may enable us to add more insights to our proposed transcriptional model. Besides the transcriptional mechanism, post-transcriptional mechanisms (alternative splicing, RNA processing, RNA silencing, et al.), post-translational modifications (protein phosphorylation, ubiquitination, sumoylation, et al.), and cross-connections among these three mechanisms will demonstrate further and superimpose complexity levels in the response to environmental changes. Future attempts at rice engineering could exploit insights from a deeper comprehension of rice–BPH interaction.

## 4. Materials and Methods

### 4.1. Plant Materials

Two *indica* rice varieties IR56 and TN with a contrasting BPH resistance nature were used in this study. Pre-germinated seeds were planted in mud beds and grown under natural light and temperature conditions in a net house. Single tillers from two-month old seedlings were transplanted to mud-filled plastic cups (diameter 12 cm, height 15 cm) and kept in a temperature-controlled greenhouse (28 ± 2 °C, 80 ± 5% relative humidity (RH)). After 10 days of transplantation, the rice plants were used for the experiments. The BPH resistance of both rice genotypes included in this study was validated in our laboratory prior to the current research.

### 4.2. Insect Materials

TN1-BPH and IR56-BPH were used to perform the BPH bioassays in this study. BPH colonies initially collected from rice fields in Hangzhou, China, were maintained on TN1 rice or IR56 rice in a climate-controlled chamber (26 ± 2 °C, 80 ± 5% RH) for more than 7 years at the China National Rice Research Institute (CNRRI). Both these populations differ in their respective virulence levels.

### 4.3. BPH Bioassays

Each rice plant with a single tiller was individually infested with 4 newly emerged adult female adults, and were confined in a transparent plastic cage (diameter 10 cm, height 60 cm) equipped with a net (with holes of diameter 0.5 mm). As per the aforementioned procedures, the two BPH populations were infested on both the IR56 and TN1 rice varieties, separately. Plants with no BPH treatment from each variety put inside a plastic cage served as a control for this experiment. Wound stress imposition was performed as described by Lu et al. [26] with a slight modification. Briefly, the lower stem portion (2 cm from the base) of each rice plant was pierced 100 times with a fine needle to induce wound stress. Plants with no piercing served as a control for this experiment. Samples were collected after 1 and 3 dpi, immediately frozen in liquid nitrogen, and stored at −80 °C for further use. The BPH assay experiments were performed with 3 independent biological replicates.

For the BPH bioassay in the chemical MAPK inhibition experiment, 40 third instars of TN1-BPH or IR56-BPH were transferred into a single pot having 10 rice plants of each rice variety. The assays were performed in a confined space by the use of a plastic cage as described above. BPH survival rates were calculated for each day of infestation by counting the number of alive BPH nymphs and dividing this number by the total number transferred. The BPH survival rates were determined for 1–8 days of infestation. The BPH assay experiments were performed with 3 independent biological replicates.

### 4.4. Chemical Treatments

Exogenous phytohormone treatments were performed as described in Nahar et al. [63] with necessary modifications. Briefly, the rice plants were individually sprayed with 100 μM MeJA (Sigma-Aldrich, St. Louis, MO, USA), 200 μM SA (Sigma-Aldrich, St. Louis, MO, USA), and 500 μM ethephon (Sigma-Aldrich, USA) until runoff to induce JA, SA, and ET treatments, respectively. Rice plants sprayed with 0.1% ethanol (*v*/*v*) served as a control for the MeJA and ethephon treatments. Rice plants sprayed with water served as a control for the SA treatments. Plant samples were collected at 1 and 3 days post-hormonal treatments, frozen immediately with liquid nitrogen, and stored at −80 °C until further use. The experiment was performed with 3 independent biological replicates.

Treatments with PD98059 and Genistein were peformed as described in Bjornson et al. [58] with necessary modifications. Breifely, the rice plants were individually sprayed with a mixture of PD98059 (Sigma-Aldrich, St. Louis, MO, USA) and Genistein (Sigma-Aldrich, St. Louis, MO, USA) of 40 µM concentration dissolved in 0.5% (*v*/*v*) dimethyl sulfoxide (DMSO) to induce MAPK inhibition. Rice plants sprayed with only 0.5% (*v*/*v*) DMSO served as a control for the experiment. The experiment was performed with 3 independent biological replicates.

### 4.5. RNA Isolation and cDNA Synthesis

Plant samples collected at the respective time points were ground to powder in a pre-chilled sterilized mortar and pestle with sufficient liquid nitrogen, and were used to isolate total RNA using the TransZol Up reagent (Transgen, Beijing, China) according to the manufacturer’s instructions. The concentration and quality of isolated RNA were determined by a NanoDrop ND-1000 spectrophotometer (Thermo-Fischer Scientific, Waltham, MA, USA) and by 1% (*w*/*v*) agarose gel electrophoresis. First-strand cDNA was synthesized from isolated RNA by using a Transcript one-step gDNA removal and cDNA synthesis supermix kit (Transgen) as per the manufacturer’s instructions.

### 4.6. Quantitative Real-Time PCR (qPCR)

Seventeen *OsMPK*s sequences were retrieved by using their RAP-DB IDs (Appendix A) and by following the nomenclature prescribed by Rohila and Yang [19]. Primers for all 17 *OsMPK*s and other defense-related genes were designed using Primer-BLAST. The qPCR was performed with a total volume of 10 μL of reaction mixture containing 5 μL of SYBR Green PCR mix (Transgen), 0.2 μL each of the specific forward and reverse primers (10 μM) (Appendix A), 0.2 μL of passive reference dye (ROX) (Thermo-Fischer Scientific, Waltham, MA, USA), 2 μL of template cDNA, and 3.4 μL of nuclease-free water. The qPCR reaction was carried out on an ABI 7500 real-time PCR system (Applied Biosystems, Foster City, CA, USA) according to the manufacturer’s instruction. Three independent biological samples for each reaction, and three technical replicates for each biological sample, were used for the qPCR analysis. The constitutively expressed housekeeping gene *OsUbq* from rice was used as an endogenous control [50]. The relative expression was evaluated using the 2^−ΔΔ*Ct*^ method [64].

### 4.7. Enzyme Activity Assays

The enzyme activities of POD (EC 1.11.1), SOD (EC 1.15.1.1), CAT (EC 1.11.1.6), and GST (EC 2.5.1.18) were determined in BPH-infested and control plants by using POD, SOD, CAT, and GST-ST kits (Nanjing Jiancheng Bioengineering Institute, Nanjing, China) by following the manufacturer’s instructions. One unit of POD activity was defined as the amount of enzyme required for a change of 0.001 in absorbance per minute. One unit of SOD activity was defined as the amount of enzyme required to inhibit 50% of the photochemical reduction of nitroblue tetrazolium. One unit of CAT activity was defined as the amount of enzyme required to change the absorbance value by 0.001 per minute at 405 nm. One unit of GST activity was defined as the amount of enzyme required to produce 1 mmol of 2,4-dinitrobenzene-glutathione conjugate per minute. All of the enzymatic assays were performed with three independent biological replicates, and each replicate included three technical replicates.

### 4.8. Statistical Analysis

The statistical analyses were carried out using Data Processing System software [65]. Data are reported as mean ± SE. For the *OsMPK* expression experiment, data were analyzed by three-way ANOVA (using rice variety, BPH population, and infestation period as the main factors), and for all other experiments, data were analyzed by two-way ANOVA followed by Tukey’s HSD post-hoc test. The statistical significance level was set for *p*-values <0.05 or 0.01.

## 5. Conclusions

In the current work, the differential roles of *OsMPK*s in two contrasting rice genotypes under BPH infestation at different virulence levels have been investigated. The findings from the study suggested that multiple factors, including rice variety, BPH population type, and infestation period have significant effects on the expression of the *OsMPK*s. Further, five *OsMPK*s (*OsMPK1*, *OsMPK3*, *OsMPK7*, *OsMPK14*, and *OsMPK16*) were found to be exclusively involved in the incompatible rice–BPH interaction. *OsMPK4*, *OsMPK8*, and *OsMPK9* exhibited induced expression when infested with cross-population BPH. An analysis of the expression of phytohormone signaling genes revealed that rice resistance to BPH might be mediated by the SA and ET signaling pathways. Further, an analysis of the transcription of phenylpropanoid biosynthesis genes suggested that BPH resistance might be realized by the boosted production of secondary metabolites. Although ROS play a vital role in insect herbivory resistance, our findings suggested that ROS accumulation in response to BPH feeding is an early response in rice. Additionally, an analysis of ROS-related enzyme activities in BPH-infested plants further strengthened this hypothesis. Lastly, we have proposed a conceptual model of interactions of IR56 rice with TN1-BPH and IR56-BPH that depicts the resistance and susceptibility reactions, respectively. Additional functional characterizations by deducing the interaction dynamics, and a loss of function analysis, will enable more insights into the role of *OsMPK*s in rice–BPH interactions.

## Figures and Tables

**Figure 1 ijms-19-04030-f001:**
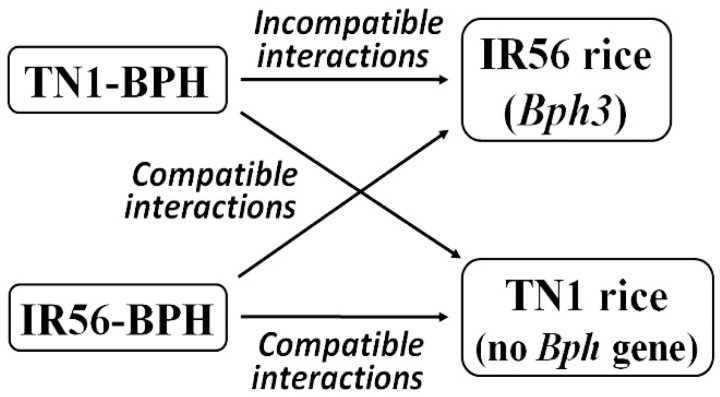
A schematic indicating the compatible and incompatible interactions among the rice varieties and brown planthopper (BPH) populations used in this study.

**Figure 2 ijms-19-04030-f002:**
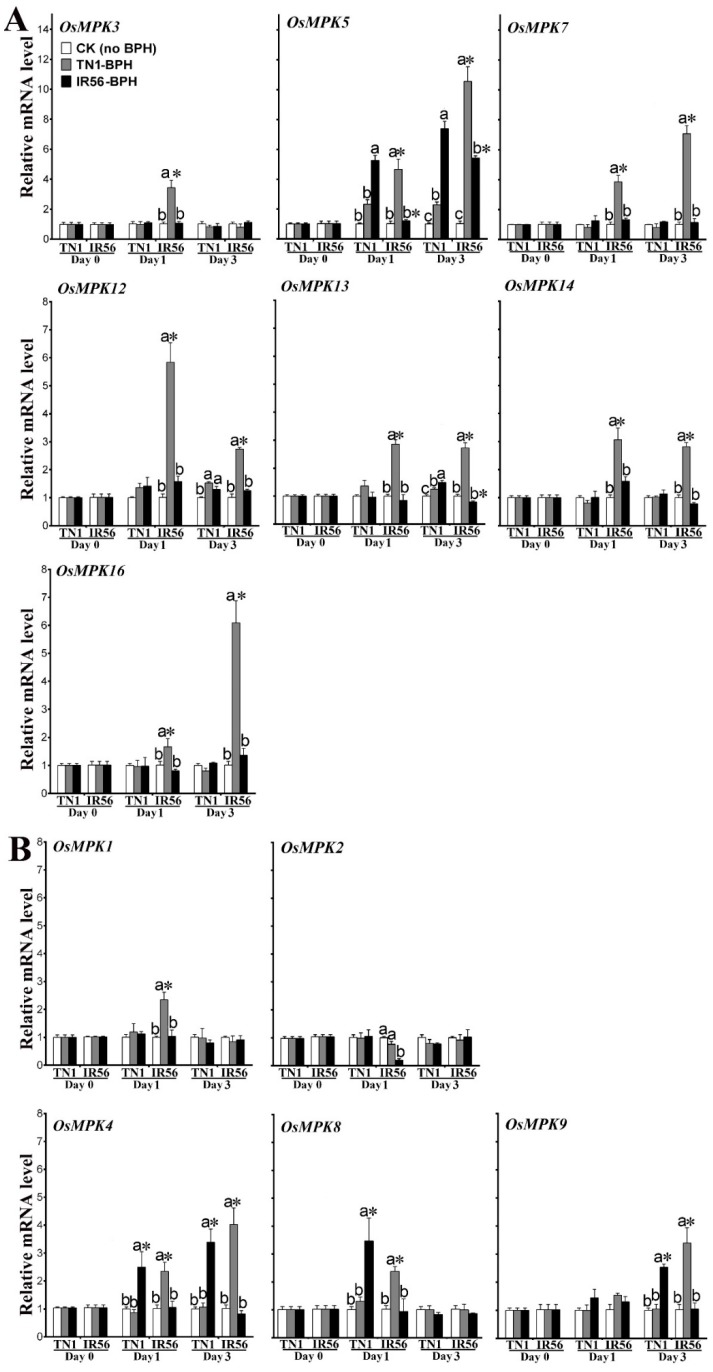
The expression profiles of the 12 *OsMPKs* showing significant induced expressions in the IR56 and TN1 rice varieties under infestation of TN1-BPH and IR56-BPH at 0–3 days post-infestation. Data are denoted as mean ± standard error (SE) and analyzed by three-way ANOVA followed by Tukey’s honestly significant difference (HSD) test. (**A**) *OsMPK*s showing significant differences in the main (rice varieties, BPH populations, and days post-infestation) and all interaction effects. (**B**) *OsMPK*s showing significant differences in any of the main effects or any of the interaction effects. * represents a significant difference with TN1 rice (*p* < 0.05); different letters above the bars represent significant differences (*p* < 0.05) among each TN1-BPH- and IR56-BPH-infested rice variety, and the control (CK, rice not infested with BPH). No significant difference among samples was not denoted by any symbol.

**Figure 3 ijms-19-04030-f003:**
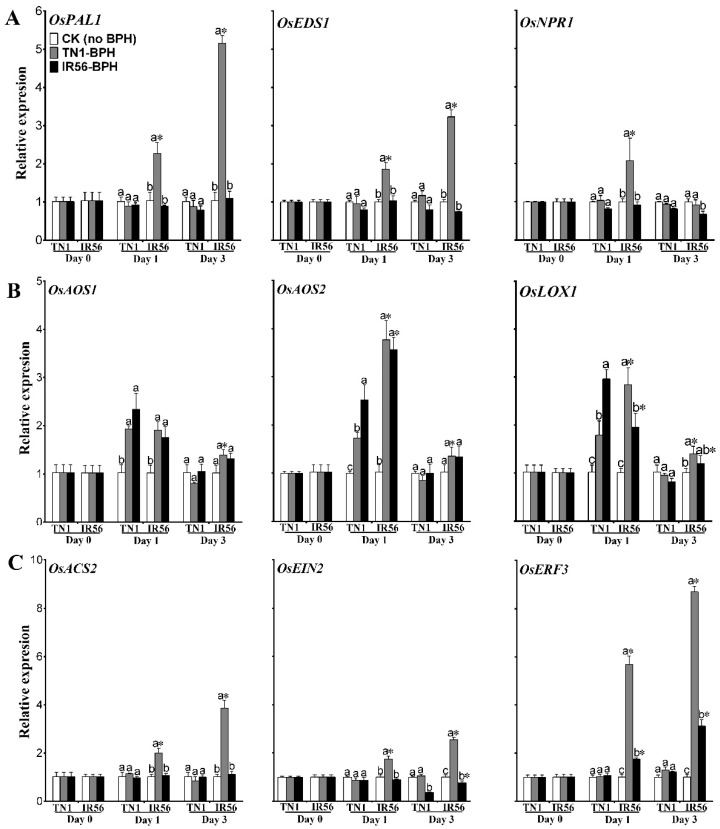
Comparative expression profiles of the phytohormone signaling genes in the IR56 and TN1 rice varieties under BPH infestation. (**A**) salicylic acid (SA) signaling genes; (**B**) jasmonic acid (JA) biosynthesis genes; (**C**) ethylene (ET) signaling genes. Data are denoted as mean ± SE and analyzed by two-way ANOVA followed by Tukey’s HSD post-hoc test. Letters above the bars indicate the significant difference (*p* < 0.05) in expression among the TN1-BPH- and IR56-BPH-infested rice varieties, and the control (rice not infested with BPH). An asterisk (*) denotes the significant difference (*p* < 0.05) in expression with respect to different rice varieties under the same infested BPH population at the specific time point. No significant difference among samples is not denoted by any symbol.

**Figure 4 ijms-19-04030-f004:**
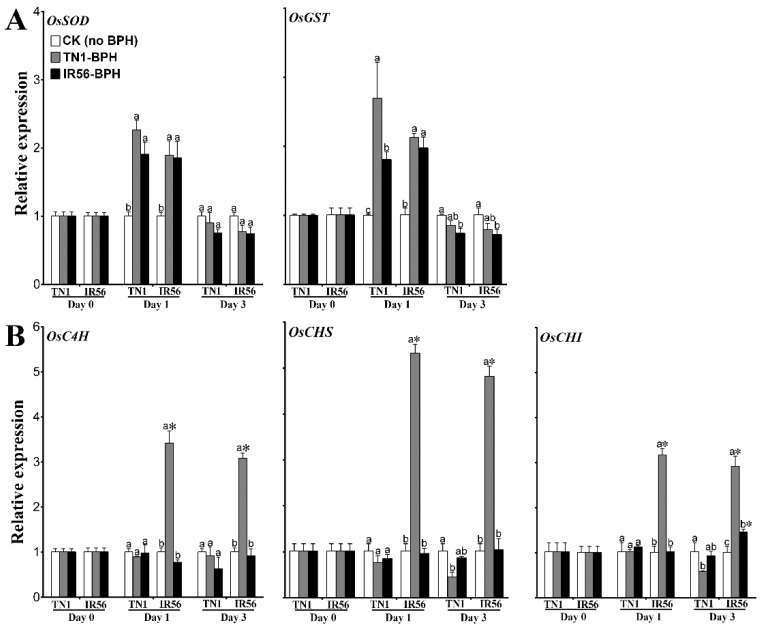
The expression profiles of phenylpropanoid biosynthesis genes and reactive oxygen species (ROS)-responsive genes in the IR56 and TN1 rice varieties under infestation by two different BPH populations. (**A**) ROS-responsive genes. (**B**) Phenylpropanoid biosynthesis genes. Data are denoted as mean ± SE and analyzed by two-way ANOVA followed by Tukey’s HSD post-hoc test. Letters above the bars indicate a significant difference (*p* < 0.05) in expression among the TN1-BPH- and IR56-BPH-infested rice varieties, and control (rice not infested with BPH). An asterisk (*) denotes a significant difference (*p* < 0.05) in expression with respect to the different rice varieties under the same infested BPH population at the specific time point. No significant difference among samples are not denoted by any symbol.

**Figure 5 ijms-19-04030-f005:**
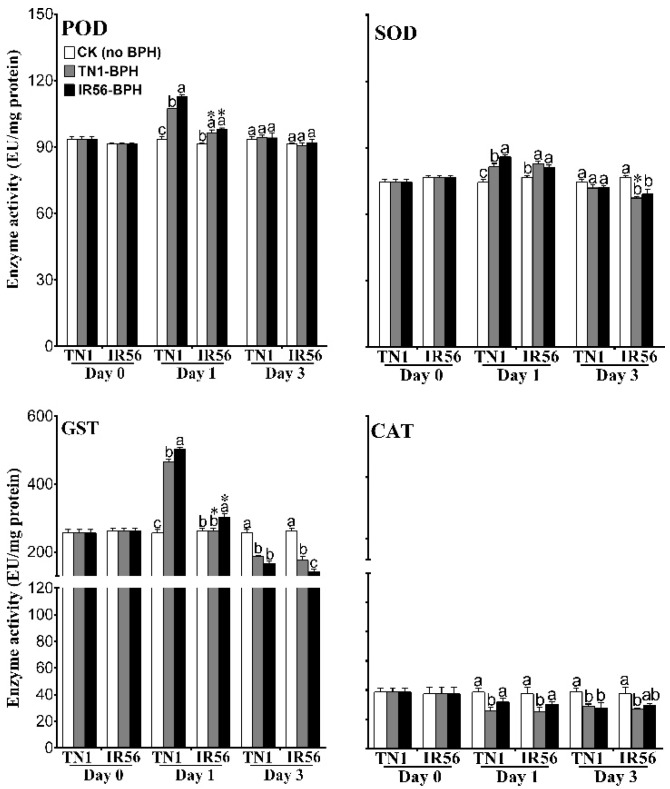
Enzyme activity of peroxidase (POD), superoxide dismutase (SOD), GST, and catalase (CAT) in the IR56 and TN1 rice varieties under infestation of the two different BPH populations. Data are denoted as mean ± SE and analyzed by two-way ANOVA followed by Tukey’s HSD post-hoc test. Letters above the bars indicate a significant difference (*p* < 0.05) in enzyme activities among TN1-BPH- and IR56-BPH-infested rice, and the control (rice not infested with BPH). An asterisk (*) denotes a significant difference (*p* < 0.05) in enzyme activity with respect to the different rice varieties under the same infested BPH population at the specific time point. No significant difference among samples are not denoted by any symbol. EU stands for enzyme unit.

**Figure 6 ijms-19-04030-f006:**
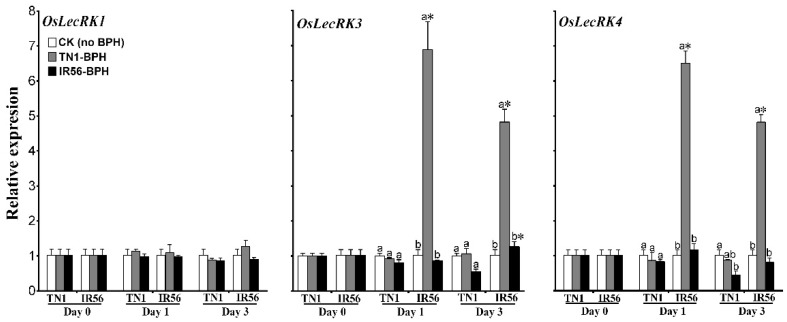
The expression patterns of *OsLecRK*s in the IR56 and TN1 rice varieties under infestation of two different BPH populations. Data are denoted as mean ± SE and analyzed by two-way ANOVA followed by Tukey’s HSD post hoc test. Letters above the bars indicate a significant difference (*p* < 0.05) in expression among TN1-BPH- and IR56-BPH-infested rice, and the control (rice not infested with BPH). An asterisk (*) denotes a significant difference (*p* < 0.05) in expression with respect to the different rice varieties under the same infested BPH population at the specific time point. No significant difference among samples are not denoted by any symbol.

**Figure 7 ijms-19-04030-f007:**
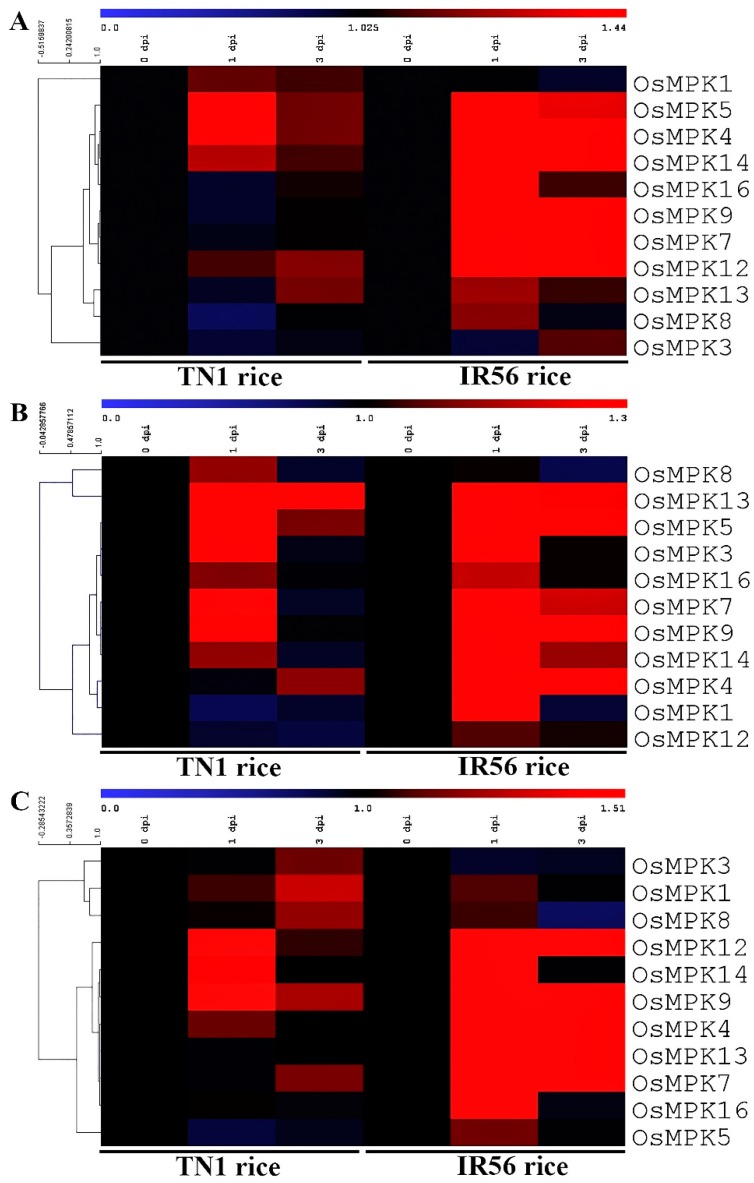
An expression analysis of the BPH-induced *OsMPKs* under exogenous treatment with different phytohormones in the IR56 and TN1 rice varieties. (**A**) Treatment of SA, (**B**) treatment of methyl jasmonate (MeJA), and (**C**) treatment of ethephon (ET). The analysis was performed based on the qPCR data of the 11 BPH-induced *OsMPK*s using the MEV program. The red color represents a positive correlation between the *OsMPK* expression and phytohormone treatment, and the blue color indicates a negative correlation between *OsMPK* expression and phytohormone treatment. The hierarchical clustering has been constructed based on the Pearson Correlation metric, and the gene order has been optimized.

**Figure 8 ijms-19-04030-f008:**
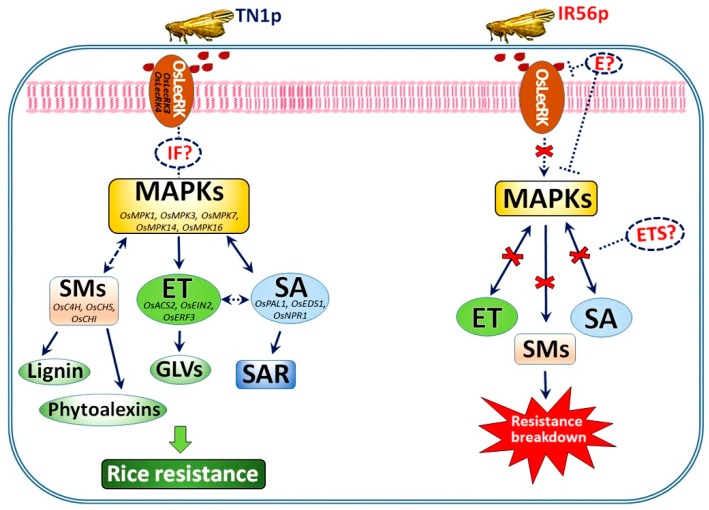
A conceptual model depicting the mechanism of the IR56 rice variety’s resistance against TN1-BPH and the breakdown of resistance in the IR56 rice variety by IR56-BPH. E: effectors; SM: secondary metabolites; ET: ethylene; SA: salicylic acid; ETS: effector-triggered susceptibility; IF: intermediate factors; GLV: green leaf volatiles; SAR: systemic acquired resistance. The solid lines indicate established regulatory connections, while the dashed lines represent a possible connection between different genes and factors.

**Table 1 ijms-19-04030-t001:** Induced expression of *OsMPK*s under different biotic stresses.

Gene Name	Biotic Stress Type
*OsMPK1*	*Magnaporthe grisea* [33]
*OsMPK2*	*M. grisea* [32]
*OsMPK3*	*M. grisea* and BPH [23]
*OsMPK4*	*M. grisea* [32] and *Chilo suppressalis* infestation [34]
*OsMPK5*	*M. grisea* [32], *C. suppressalis* [22], and BPH infestation [28]
*OsMPK6*	*C. suppressalis* infestation [30]
*OsMPK7*	*M. grisea* [32] and *Xanthomonas oryzae* [35]
*OsMPK8*	*M. grisea* [32]
*OsMPK12*	Fungal elicitor chitosan [36], *M. grisea* [32], and BPH infestation [28]
*OsMPK13*	*M. grisea* [32] and BPH infestation [28]
*OsMPK15*	*M. grisea* [32]
*OsMPK17*	*M. grisea* [32] and BPH infestation [28]

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
