# Peer review of "Differential Responses of OsMPKs in IR56 Rice to Two BPH Populations of Different Virulence Levels"

_ijms, 2018, doi:10.3390/ijms19124030_

Round 1

Reviewer 1 Report

The authors explored the role of all 17 rice MAPKs in the interactions of two different brown planthopper (BPH) genotypes with a resistant and a susceptible rice variety. They used  transcription profiling (qPCR) of MAPK-, defense-, and hormone marker genes as well as enzymatic assays for antioxidant enzymes to determine correlations between MAPKs and plant defenses against BPH. Significant correlations were determined using ANOVA tests.

Three-way ANOVA tests determined whether MPK expression is dependent on rice variety, BPH genotype, and infestation period (0-3 days). ANOVA data identified five MAPK genes that may contribute to the incompatibility of TN1-BPH and IR56-rice. In addition, the authors identified three SA-, three Ethylene-, and three phenylpropanoid-related marker genes as upregulated in the incompatible interaction between TN1-BPH and IR56-rice, but not in incompatible interactions, whereas JA biosynthesis genes and antioxidant enzyme genes and activities are upregulated in both interactions. Also, more MAPKs are upregulated by external application of SA, MeJA, and ethephon (ethylene) in the IR56 variety than in the TN1 variety. (The IR56 variety is resistant to TN1-BPH.) An interesting discovery was that LecRK genes are suppressed in compatible interactions, indicating that BPH employs strategies like effector-triggered susceptibility, similar to plant-pathogen interactions.

This study provides a wealth of MAPK gene expression data. It also succeeds in correlating MAPK gene expression, hormone and stress marker gene expression, and hormone treatments with compatibility and incompatibility of the BPH-rice interaction. However, the authors did not determine the mechanistic basis for incompatibility between TN1-BPH and IR56-rice, but rather identified correlations, not causal effects. What are the implications for resistance or susceptibility? This is a limitation of any study focused on gene expression alone. But it provides a basis for future studies aimed at targeting interesting MAPK genes and studying the role of the protein for rice defenses against BPH.

There are a number of concerns that should be addressed.

Abstract:

The conclusion that BPH resistance may not be regulated by JA should be toned down. After all, JA biosynthetic genes were upregulated in response to BPH. Even though this is not specific to incompatible interactions, JA may still be required for resistance against BPH.

Introduction:

The terminology is very confusing. Why did the authors give BPH and rice genotypes (or biotypes) the same name, only distinguishing them by ‘population’ for the BPH and ‘variety’ for the host plant genotypes. It would be more clear to associate ‘BPH’ and ‘rice’ with the biotype or variety, such as TN1-BPH and TN1-rice. In plant pathology, gene-for-gene interactions often have matching names for both avirulence and resistance genes.

Results:

- No figure legends were provided to reviewers for supplementary figures and tables. Without those, the data are difficult to evaluate. It is not clear how Fig. S1 and Table S1 correspond to each other, which was implied in the Results text. Apparently, Figure S1 is similar to Figure 1, but that should be stated somewhere and the legend must be included. The same is true for Figure S2.

- In the text (109), MPK14 was included in the group with no significant effects, but Table S1 shows the opposite. MPK14 was then (line 110) also included in the ‘significant’ group, so it is probably an error.

- The quality of Figures 1 and 2 is poor, x-axis annotation should be improved (bigger font size). For ease of use, x-axis should be annotated separately for each panel (A,B,C). I realize that the reviewer version may have a lower resolution, but the font size may still be a problem in the final version.

- The Results text refers to Fig S1 (starting line 164), but this figure was provided as Fig. S2 (pricking experiments).

- Gene expressions in Fig S2 are shown as significant that are only slightly different in terms of ‘relative expression’. E.g. the Results text (line 170) states: “significant effect of rice varieties (F1,12  = 4.76, P  = 0.049) and dpi (F2,12  = 5.62, P  = 0.019) were found on OsMPK6 expression.” While the differences may be statistically significant, do the authors imply that such small changes in gene expression have consequences for resistance/susceptibility? Accounting for posttranscriptional and posttranslational regulatory mechanisms, it is not possible to derive a meaningful difference for resistance from the data as shown. This should be discussed.

- For the pricking experiments, it is not always clear to what figure or table the data mentioned in the text refer.

- Line 178: OsMPK13 and OsMPK8 were mixed up.

- Figure 2:  It would be better to call the JA genes ‘biosynthesis’ genes, not signaling genes. Signaling genes would function after JA or JA-Ile has been synthesized.

- Line 227f: “significant up-regulation of JA signaling genes at 1 dpi, but not at 3 dpi, and during both compatible and incompatible rice-BPH interactions suggest that JA signaling might be involved in early response  to the BPH feeding in rice, rather than BPH resistance in rice.” Statements like that should be toned down. JA biosynthesis genes are known to be upregulated very quickly after wounding or insect attack (within minutes) and mRNA levels go down within a few hours. Upon a 3-day attack by BPH, that may be different, but in order to make a statement about JA signaling in this context, it would be important to measure JA levels, not only gene expression.

- Figure 4: POD and SOD activities at day 1 are said to be elevated as compared to days 0 and 3. Do the authors have evidence that these very small differences have any functional consequence? Same is true for some other data, e.g. for CAT. On the other hand, in that experiment, GST in the TN1 variety clearly stands out.

- Line 263: “significant decrease in the enzyme activity of POD … at 3 dpi.” In Figure 4, that is not shown for POD, only for SOD; but again, the differences seem to be minute.

- Fig. 4 needs to be correlated with Tables S3-5. Those tables were not mentioned in the text.

- What is shown by the colors in Fig. 4 in terms of value? Relative gene expression?

Discussion

- There are more language errors in the discussion than elsewhere in the text. The entire manuscript should be proof-read. Although, the writing is generally adequate.

- first paragraph. It would be helpful to provide a table summarizing the text, e.g. listing all OsMAPKs and what type of biotic stressor is inducing their gene expression or activity, plus references.

- Line 381. The fact that ET- and SA-related genes are both upregulated by BPH does not demonstrate synergism between SA and ethylene. This can only be determined by testing SA and ethylene mutants.

- Line 400 and 408: “MAPK-mediated secondary metabolite production”. This is an overstatement. MAPKs may or may not be involved. The data presented here only show correlations.

- Last paragraph: HAMP/DAMP activation of LecRKs is a reasonable speculation. However, is it known that feeding of BPH involves degradation of the plant cell walls resulting in the generation of oligogalacturonide DAMPs. This would require the action of a polygalacturonase. The suggested mechanism of OsMPK activation by LecRKs is unclear. Plant MAPKKKs, let alone MAPKs, are not known to be directly activated by membrane-bound receptors, and neither by calcium ions (calcium fluxes often coincide with MAPK activation).

- 450: Do the authors suggest that MAPKs regulate JA synthesis, which then would result in upregulation of ethylene and SA? That does not seem to be likely based on the available data presented by the authors and in the literature.

- Figure 7: Does this imply a role of SAR in resistance of rice to BPH? The text mentions callose and proteinase inhibitors, which are probably more local defenses.

References:

- not listing them alphabetically or numbered made this review harder.

- Hu et al, Plant Phys 2011 (DOI: https://doi.org/10.1104/pp.111.174334) was not referenced. They showed BPH-induced upregulation of OsMPK5,12, 13, and 17.

Author Response

The authors explored the role of all 17 rice MAPKs in the interactions of two different brown planthopper (BPH) genotypes with a resistant and a susceptible rice variety. They used  transcription profiling (qPCR) of MAPK-, defense-, and hormone marker genes as well as enzymatic assays for antioxidant enzymes to determine correlations between MAPKs and plant defenses against BPH. Significant correlations were determined using ANOVA tests.

Three-way ANOVA tests determined whether MPK expression is dependent on rice variety, BPH genotype, and infestation period (0-3 days). ANOVA data identified five MAPK genes that may contribute to the incompatibility of TN1-BPH and IR56-rice. In addition, the authors identified three SA-, three Ethylene-, and three phenylpropanoid-related marker genes as upregulated in the incompatible interaction between TN1-BPH and IR56-rice, but not in incompatible interactions, whereas JA biosynthesis genes and antioxidant enzyme genes and activities are upregulated in both interactions. Also, more MAPKs are upregulated by external application of SA, MeJA, and ethephon (ethylene) in the IR56 variety than in the TN1 variety. (The IR56 variety is resistant to TN1-BPH.) An interesting discovery was that LecRK genes are suppressed in compatible interactions, indicating that BPH employs strategies like effector-triggered susceptibility, similar to plant-pathogen interactions.

This study provides a wealth of MAPK gene expression data. It also succeeds in correlating MAPK gene expression, hormone and stress marker gene expression, and hormone treatments with compatibility and incompatibility of the BPH-rice interaction. However, the authors did not determine the mechanistic basis for incompatibility between TN1-BPH and IR56-rice, but rather identified correlations, not causal effects. What are the implications for resistance or susceptibility? This is a limitation of any study focused on gene expression alone. But it provides a basis for future studies aimed at targeting interesting MAPK genes and studying the role of the protein for rice defenses against BPH.

There are a number of concerns that should be addressed.

Abstract:

The conclusion that BPH resistance may not be regulated by JA should be toned down. After all, JA biosynthetic genes were upregulated in response to BPH. Even though this is not specific to incompatible interactions, JA may still be required for resistance against BPH.

Response: Sentences implying such conclusions have been toned down and revised.

Introduction:

The terminology is very confusing. Why did the authors give BPH and rice genotypes (or biotypes) the same name, only distinguishing them by ‘population’ for the BPH and ‘variety’ for the host plant genotypes. It would be more clear to associate ‘BPH’ and ‘rice’ with the biotype or variety, such as TN1-BPH and TN1-rice. In plant pathology, gene-for-gene interactions often have matching names for both avirulence and resistance genes.

Response: The terminology has been updated throughout the manuscript as suggested by the reviewer. BPH populations have been renamed as TN1-BPH or IR56-BPH, whereas, TN1 or IR56 varieties have been replaced with TN1 or IR56 rice.

Results

1.      No figure legends were provided to reviewers for supplementary figures and tables. Without those, the data are difficult to evaluate. It is not clear how Fig. S1 and Table S1 correspond to each other, which was implied in the Results text. Apparently, Figure S1 is similar to Figure 1, but that should be stated somewhere and the legend must be included. The same is true for Figure S2.

Response: Please accept our apologies for the inconvenience. All the supplementary figures and tables are now have been provided with proper legends. Further, Figure S1 is actually a part of Figure 1 (now figure 2 as per the revised manuscript) (showing the non-significant MAPK expressions), but due to space limitations we have put it as a supplementary figure. Legends for Figure S1 and S2 have been updated, and cited in the text.

2.      In the text (109), MPK14 was included in the group with no significant effects, but Table S1 shows the opposite. MPK14 was then (line 110) also included in the ‘significant’ group, so it is probably an error.

Response: Our apologies for the typo error. The MPK14 was included in the group with significant effects. The sentences has been correctly revised accordingly.

3.      The quality of Figures 1 and 2 is poor, x-axis annotation should be improved (bigger font size). For ease of use, x-axis should be annotated separately for each panel (A,B,C). I realize that the reviewer version may have a lower resolution, but the font size may still be a problem in the final version.

Response: The quality of all the figures have been improved. Font sizes of all the figures were increased for better visibility. As suggested by the reviewer, the x-axis annotations have been placed for each figure for the ease of use.

4.      The Results text refers to Fig S1 (starting line 164), but this figure was provided as Fig. S2 (pricking experiments).

Response: The sentence has been revised correctly.

5.      Gene expressions in Fig S2 are shown as significant that are only slightly different in terms of ‘relative expression’. E.g. the Results text (line 170) states: “significant effect of rice varieties (F1,12  = 4.76, P  = 0.049) and dpi (F2,12  = 5.62, P  = 0.019) were found on OsMPK6 expression.” While the differences may be statistically significant, do the authors imply that such small changes in gene expression have consequences for resistance/susceptibility? Accounting for posttranscriptional and posttranslational regulatory mechanisms, it is not possible to derive a meaningful difference for resistance from the data as shown. This should be discussed.

Response: In the manuscript, the transcriptional changes of OsMPKs in IR56 rice to BPH infestation and mechanical wounding were revealed. In plant, the molecular response to abiotic or biotic stresses has been often considered as a complex process mainly based on the modulation of transcriptional activity of stress-related genes. Moreover, the posttranscriptional mechanisms (alternative splicing,RNA processing, RNA silencing et al), posttranslational modifications (protein phosphorylation, ubiquitination, sumoylation et al), and cross-connections among these three mechanisms demonstrated further and superimposed complexity levels in the response to environmental changes. Future attempts of rice engineering could exploit insights from a deeper comprehension of rice-insect interaction. We added some discussion.

 6.      For the pricking experiments, it is not always clear to what figure or table the data mentioned in the text refer.

Response: Such ambiguities have been carefully removed and proper figures have been cited in the text for reference.

7.      Line 178: OsMPK13 and OsMPK8 were mixed up.

Response: The sentence has been corrected.

8.      Figure 2:  It would be better to call the JA genes ‘biosynthesis’ genes, not signaling genes. Signaling genes would function after JA or JA-Ile has been synthesized.

Response: As suggested by the reviewer, the term signaling genes in reference to JA signaling have been replaced with biosynthesis genes.

9.      Line 227f: “significant up-regulation of JA signaling genes at 1 dpi, but not at 3 dpi, and during both compatible and incompatible rice-BPH interactions suggest that JA signaling might be involved in early response  to the BPH feeding in rice, rather than BPH resistance in rice.” Statements like that should be toned down. JA biosynthesis genes are known to be upregulated very quickly after wounding or insect attack (within minutes) and mRNA levels go down within a few hours. Upon a 3-day attack by BPH, that may be different, but in order to make a statement about JA signaling in this context, it would be important to measure JA levels, not only gene expression.

Response: Such sentences have been toned down, and rephrased carefully.

10.  Figure 4: POD and SOD activities at day 1 are said to be elevated as compared to days 0 and 3. Do the authors have evidence that these very small differences have any functional consequence? Same is true for some other data, e.g. for CAT. On the other hand, in that experiment, GST in the TN1 variety clearly stands out.

Response: Here we have determined the effect of BPH infestation on ROS generation or subsequent signaling only by calculating the selected ROS-related enzyme activities. However, the functional consequences of the small differences in their activities were not determined.

11.  Line 263: “significant decrease in the enzyme activity of POD … at 3 dpi.” In Figure 4, that is not shown for POD, only for SOD; but again, the differences seem to be minute.

Response: The decrease in the enzyme activity was observed only in SOD instead of POD and SOD. The sentences has been corrected accordingly. Further, Figure 4 (now figure 5 as in the revised version of the manuscript) has been correlated with Table S2, and mentioned in the text.

12.  Fig. 4 needs to be correlated with Tables S3-5. Those tables were not mentioned in the text.

Response: We believe the reviewer is mentioning about figure 6 here. Figure 6 (now figure 7 as in the revised version of the manuscript) has been correlated with Table S3-5, and mentioned in the text as suggested by the reviewer.

13.  What is shown by the colors in Fig. 4 in terms of value? Relative gene expression?

Response: The colour and intensity of the boxes is used to represent relative expressions (not absolute values) of the genes.

Discussion

1.      There are more language errors in the discussion than elsewhere in the text. The entire manuscript should be proof-read. Although, the writing is generally adequate.

Response: The manuscript has been proof read by an expert for English improvements.

2.      First paragraph. It would be helpful to provide a table summarizing the text, e.g. listing all OsMAPKs and what type of biotic stressor is inducing their gene expression or activity, plus references.

Response: As suggested by the reviewer, a table listing the induced responses of different OsMPKs under different biotic stresses has been provided with references.

3.      Line 381. The fact that ET- and SA-related genes are both upregulated by BPH does not demonstrate synergism between SA and ethylene. This can only be determined by testing SA and ethylene mutants.

Response: The sentence has been removed as no experimental evidence is there to support the hypothesis. The discussion portion for ET and SA-related gene expressions have been revised, keeping the key inference indicated by the reviewer in mind.

 4.      Line 400 and 408: “MAPK-mediated secondary metabolite production”. This is an overstatement. MAPKs may or may not be involved. The data presented here only show correlations.

Response: The sentence has been rephrased.

5.      Last paragraph: HAMP/DAMP activation of LecRKs is a reasonable speculation. However, is it known that feeding of BPH involves degradation of the plant cell walls resulting in the generation of oligogalacturonide DAMPs. This would require the action of a polygalacturonase. The suggested mechanism of OsMPK activation by LecRKs is unclear. Plant MAPKKKs, let alone MAPKs, are not known to be directly activated by membrane-bound receptors, and neither by calcium ions (calcium fluxes often coincide with MAPK activation).

Response:In the proposed model, we don’t claim the activation of MAPK directly by the LecRKs or Ca2+ ions. As pointed by the reviewer, the paragraph has been rephrased and the proposed hypothesis has been revised.

6.      Do the authors suggest that MAPKs regulate JA synthesis, which then would result in upregulation of ethylene and SA? That does not seem to be likely based on the available data presented by the authors and in the literature.

Response: As for the current study, we don’t suggest the correlation between OsMPKs and JA synthesis. Rather, our hypothesis is SA and ET play important role in BPH resistance mechanism. The results obtained in our study is in accordance to some of the previously reported studies on rice-insect interactions, as discussed in the text.

7.      Figure 7: Does this imply a role of SAR in resistance of rice to BPH? The text mentions callose and proteinase inhibitors, which are probably more local defenses.

Response: Role of SAR in plant resistance is critical, and can’t be ruled out in rice-BPH interactions. Upregulated expression of SA signaling genes (in our study) and deposition of callose (from previous studies) suggest the potential role of SAR in rice resistance against BPH infestation. Here, we have summarized and proposed a model for rice-BPH interactions based on our findings and the available literature.

References

1.      Not listing them alphabetically or numbered made this review harder.

Response: Kindly accept our apologies for the inconvenience. References have been arranged numerically in the revised version of the manuscript.

2.      Hu et al, Plant Phys 2011 (DOI: https://doi.org/10.1104/pp.111.174334) was not referenced. They showed BPH-induced upregulation of OsMPK5, 12, 13, and 17.

Response: Thank you for your excellent remark on the literature we’ve missed. The same has been referenced and included in the text for discussion.

Reviewer 2 Report

This paper describes the transcriptional patterns of several different MAPKinases,  hormone-responsive, ROS-signaling and secondary metabolite genes in Oryza sativa varieties under different "strains" of brown plant hoppers.  The methods are scientifically sound and overall, the paper has merit but needs some additional descriptions and clarifications.

Lines 133-137 are initially confusing and need to be clarified.  I suggest putting a schematic or graphic in the introduction that describes your hypothesis for virulent/avirulent or compatible/incompatible reactions for the rice varieties with the insects.  

Most plant-pathogen interaction studies focus on virulent vs. virulence as being a gene-for-gene resistance interaction.  Although it is reference in Zheng et al. 2016, could you briefly describe this for clarity in this publication?

Figure 7: There is no reference in the paper to the hypersensitive response/acquired resistance  or to what an incompatible response is.  There should be a few short sentences describing this as it is the "ultimate end" in the Figure 7. 

Line: 40 You mention that BPH can be vectors for stunt viruses, but never allude to whether this could be the cause of the compatible reactions in IR156/IR156 treatment.  In lines 355-366 You note that it could be insect saliva or effector-mediated susceptibility.  Have you done PCR to  test for the presence of either stunt virus strain in your BPH populations?  You should discuss why this was or was not done in the manuscript or whether this could even be a source of the compatible interaction. 

Lines: 262, 287 (vs. Line 301).  Use the word "rice" instead of varieties.  Thus, all references to rice should be TN1 rice or IR156 Rice.  You use the words "TN1 or IR156 populations" (for example Lines 142-143) to describe the insects in numerous occasions (too numerous to list individually).  Perhaps a better term would be TN1-fed insect populations or TN1-native populations.  This way the reader won't get confused about "varieties and populations" and which is the plant vs. the insect that you are referring to. 

Line 187:  You need a reference after this first sentence. 

Line 432: Interplay, not interplays

Figures 1-4:  Please put  X-axis labels on ALL graphs, not just down at the bottom.  It is hard to follow up and I had to draw lines up to keep remembering the different treatment groups. 

Figure 6 is a little confusing.  Typically green in a heat map or microarray means up-regulated (not negative) while red is down-regulated.  If you don't want to switch the red/green, then I suggest making the green a bluish color.  First, those who suffer from red/green colorblindness will not be able to read your graph and it won't conflate your heat map with typical microarray coloration. 

Lines 483-484:  Include in the supplementary materials how you confirmed BPH resistance in the rice varieties. 

Author Response

1.      Lines 133-137 are initially confusing and need to be clarified.  I suggest putting a schematic or graphic in the introduction that describes your hypothesis for virulent/avirulent or compatible/incompatible reactions for the rice varieties with the insects. 

Response: Line 133-137 has been revised to make the context clearer. A schematic indicating the compatible and incompatible interactions among rice varieties and BPH populations has been incorporated in the introduction section as suggested by the reviewer.

2.      Most plant-pathogen interaction studies focus on virulent vs. virulence as being a gene-for-gene resistance interaction.  Although it is reference in Zheng et al. 2016, could you briefly describe this for clarity in this publication?

Response: As suggested by the reviewer, a brief description of the outcomes from the study by Zheng et al. 2016, in reference to IR56 rice resistance breakdown by IR56-BPH have been added to the introduction part to provide more clarity.

3.      Figure 7: There is no reference in the paper to the hypersensitive response/acquired resistance or to what an incompatible response is.  There should be a few short sentences describing this as it is the "ultimate end" in the Figure 7.

Response: As suggested by the reviewer, this portion has been revised including information on compatible and incompatible interactions in connection to HR and other acquired resistance responses.

4.      Line: 40 You mention that BPH can be vectors for stunt viruses, but never allude to whether this could be the cause of the compatible reactions in IR156/IR156 treatment.  In lines 355-366 You note that it could be insect saliva or effector-mediated susceptibility.  Have you done PCR to test for the presence of either stunt virus strain in your BPH populations?  You should discuss why this was or was not done in the manuscript or whether this could even be a source of the compatible interaction.

Response: The BPH populations are maintained very carefully, and routinely checked for loss of virulence due to propagation, and infections such as viruses. The BPH used in the study have been confirmed to be free from such adversities, before commencing the experiments.

5.      Lines: 262, 287 (vs. Line 301).  Use the word "rice" instead of varieties.  Thus, all references to rice should be TN1 rice or IR156 Rice.  You use the words "TN1 or IR156 populations" (for example Lines 142-143) to describe the insects in numerous occasions (too numerous to list individually).  Perhaps a better term would be TN1-fed insect populations or TN1-native populations.  This way the reader won't get confused about "varieties and populations" and which is the plant vs. the insect that you are referring to.

Response: The nomenclature has been revised throughout the manuscript. IR56 or TN1 varieties have been replaced with IR56 or TN1 rice. Similarly, IR56 or TN1 population have been replaced by IR56-BPH or TN1-BPH (as also suggested by reviewer 1).

6.      Line 187:  You need a reference after this first sentence.

Response: A reference has been added.

7.      Line 432: Interplay, not interplays

Response: Corrected as suggested by the reviewer.

8.      Figures 1-4:  Please put  X-axis labels on ALL graphs, not just down at the bottom.  It is hard to follow up and I had to draw lines up to keep remembering the different treatment groups.

Response: Kindly accept our apologies for the inconvenience. We have revised all the figures by providing X-axis annotations and increasing the font sizes to enhance the visibility and ease of use.

9.      Figure 6 is a little confusing.  Typically green in a heat map or microarray means up-regulated (not negative) while red is down-regulated.  If you don't want to switch the red/green, then I suggest making the green a bluish color.  First, those who suffer from red/green colorblindness will not be able to read your graph and it won't conflate your heat map with typical microarray coloration.

Response: As per the reviewer’s suggestion, the greens in the heat map have been replaced by blue color. 

10.  Lines 483-484:  Include in the supplementary materials how you confirmed BPH resistance in the rice varieties.

Response: The individual reactions of the rice varieties (TN1 or IR56) to each BPH population type (TN1-BPH or IR56-BPH) have already been studied and published in our lab (Zheng et al. 2016). We have included the reference in text as well as in the reference list.

Reviewer 3 Report

The brown planthopper is an important pest of rice and resistant breeding has been a successful protection strategy. The molecular mechanisms of resistance involve PTI and ETI via MPK and hormone signalling (reviewed in Ling 2016 Plant Cell Reports). In recent year, resistant mediated by Bph genes is overcome by the planthopper.  In the present manuscript, the brown planthopper-rice interaction of such a population is characterized transcriptionally with a focus on MPK, hormone and ROS signalling.

While understanding how plant resistance is broken is a highly interesting topic, I do not see this question answered in the data presented in this manuscript. The dataset comprises the transcriptional response of virulent and avirulent populations in susceptible and resistant cultivars in a 3-day timecourse. Figures 1-3, 5 show the response of different genes. As expected an overall general trend is the induction of several of these markers in the resistant cultivar infected with the avirulent population. In addition, plant responses to hormone treatment and enzymatic assays for ROS enzymes are shown. From this the authors draw a model placing some genes into the known signalling cascades.

Overall, the limited dataset and the very descriptive style of presenting these data did not convey to me any novelty.

Nevertheless the dataset in itself is interesting for the community, but should be published combined with some kind of functional data on this important interaction to really understand the resistance breakdown.

Author Response

The brown planthopper is an important pest of rice and resistant breeding has been a successful protection strategy. The molecular mechanisms of resistance involve PTI and ETI via MPK and hormone signalling (reviewed in Ling 2016 Plant Cell Reports). In recent year, resistant mediated by Bph genes is overcome by the planthopper. In the present manuscript, the brown planthopper-rice interaction of such a population is characterized transcriptionally with a focus on MPK, hormone and ROS signalling.

While understanding how plant resistance is broken is a highly interesting topic, I do not see this question answered in the data presented in this manuscript. The dataset comprises the transcriptional response of virulent and avirulent populations in susceptible and resistant cultivars in a 3-day timecourse. Figures 1-3, 5 show the response of different genes. As expected an overall general trend is the induction of several of these markers in the resistant cultivar infected with the avirulent population. In addition, plant responses to hormone treatment and enzymatic assays for ROS enzymes are shown. From this the authors draw a model placing some genes into the known signaling cascades.

Overall, the limited dataset and the very descriptive style of presenting these data did not convey to me any novelty.

Nevertheless the dataset in itself is interesting for the community, but should be published combined with some kind of functional data on this important interaction to really understand the resistance breakdown.

Response: Thanks for your suggestion. In this manuscript, the transcriptional response of virulent and avirulent populations in susceptible and resistant cultivars was revealed, and should be interesting for the community. The functional analysis of BPH effectors and rice MAPK-relative genes is in progress in our lab and the results may enable us to understand the resistance breakdown.

Round 2

Reviewer 3 Report

In the revised version, I really liked the scheme in figure 1 which illustrates the types of interactions. Moreover, the changes in the text make it a lot easier to understand and the description of the transcriptional changes is clearly presented and well-written.

I agree that the dataset is interesting and relevant for the community. However, I still miss the functional data. I understand that a full characterization is way beyond the scope of this manuscript, but I find it essential to support the description of the transcriptional response with some additional data that point into the direction how this dataset contributes to understanding the resistance breakdown. For example, does the use of MPK-inhibitors interfere with resistance and/or increase susceptibility in the 4 interactions. This would be an experimental set-up, which does not require generation of MPK-mutants. However, if the authors already have such mutants at hand that carry the resistance gene, presenting the outcome of such infections would much strengthen the manuscript.

Author Response

In the revised version, I really liked the scheme in figure 1 which illustrates the types of interactions. Moreover, the changes in the text make it a lot easier to understand and the description of the transcriptional changes is clearly presented and well-written.

I agree that the dataset is interesting and relevant for the community. However, I still miss the functional data. I understand that a full characterization is way beyond the scope of this manuscript, but I find it essential to support the description of the transcriptional response with some additional data that point into the direction how this dataset contributes to understanding the resistance breakdown. For example, does the use of MPK-inhibitors interfere with resistance and/or increase susceptibility in the 4 interactions. This would be an experimental set-up, which does not require generation of MPK-mutants. However, if the authors already have such mutants at hand that carry the resistance gene, presenting the outcome of such infections would much strengthen the manuscript.

Response: Thank you for your positive comments on the revised version of the manuscript, and your critical criticism for the further improvement of the manuscript. Regarding the functionality assessment of the MAPKs by MAPK inhibition, to our best knowledge no specific MAPK-chemical inhibitors are available for our identified candidate OsMPKs (five OsMPKs those exhibited induced expression exclusively during the incompatible rice-BPH interaction). However, we have already performed the chemical MAPK-inhibitions (non-specific, broad-spectrum) by using the mixture of two chemical inhibitors, including PD98059 and Genistein. Nonetheless, the results of these experiments were found to be inconclusive, so were not included in the previous version of the manuscript. In this revision, we have discussed the findings from the chemical MAPK-inhibition experiment. Also, we have revised the material and method portion by adding the information about the treatment of PD98059 and Genistein, and the BPH bioassays on treated rice varieties. Briefly, the treatment of the chemical MAPK inhibitors resulted in a transient down-regulation in OsMPK expressions, however, no significance difference in the BPH performances were found in between inhibitor-treated and control plants. On the other hand, plants repeatedly treated with inhibitors to prolong the transient MAPK-inhibition, showed fatal drying of leaves and were dead in 5-8 days of the treatment. Therefore, the selective mutation of the candidate MAPK genes can be a suitable option to elucidate their functionality.

At the current stage, we are making specific CRISPR-vectors for the candidate MAPKs for the further studies. Alongside, we have made CRISPR-OsLecRK3 and CRISPR-OsLecRK4 vectors ready for the mutation studies. We believe, the outcomes from the planned mutation assays will add insights to the functional part of this particular experiment.